# ADAPTING RETRIEVAL MODELS TO TASK-SPECIFIC GOALS USING REINFORCEMENT LEARNING

## ABSTRACT

Given an input query, retrieval models are trained using user feedback data (e.g., click data) to output a ranked list of items. However, it is difficult to optimize task-specific goals using supervised learning because the goals often correspond to non-differentiable losses. For example, we may want to optimize recall or novelty of the top-k items for a recommendation task or optimize accuracy of a black-box large language model (LLM) for the retrieval-augmented generation task. To optimize arbitrary task-specific losses, we propose a reinforcement learning-based framework that applies to any pretrained retrieval model. Specifically, our solution uses policy gradient and addresses the key challenge of large action spaces by reduction to a binary action space, given both the query and the retrieved item. Our formulation also allows for exploration based on auxiliary retrieval models. We empirically evaluate the proposed algorithm on improving recall for a query-ad retrieval task on two datasets with 4K and 1.9M actions respectively. We also show the benefit of the proposed algorithm on improving a custom metric—novelty of the retrieved items w.r.t. existing algorithms—for a commercial search engine.

## 1 INTRODUCTION

Given an input query, the retrieval problem is to fetch a ranked list of top-k items based on a task-specific goal. For example, in search and recommendation systems, semantic relevance is a common goal and retrieval models are trained to output top-k relevant items to a user's query (Guo et al., 2022). Recently, retrieval models have also been used for selecting in-context examples for a large language model, a task known as retrieval-augmented generation (Lewis et al., 2020). A common way to train such retrieval models is to use supervised learning on user feedback data such as clicks. For instance, losses such as contrastive learning encourage representations of positive query-item pairs to be closer to each other than negative (or random) query-item pairs (Gao et al., 2021).

However, the supervised learning losses often do not match the stated goals of a retrieval task. For instance, the contrastive loss encourages the model representations to be closer for relevant query-item pairs (and hence relevant items may be ranked higher), but it does not directly optimize for the quality of the top-$k$ ranked list of items. Since the output of retrieval models is a top-$k$ ranked list, the training goal ideally should be defined wrt. properties of the top-k list. For example, one may wish to optimize the precision or recall of the returned set (Harman, 2011). As another example, in a real-world recommendation system, one may be interested in the novelty of the top-$k$ items compared to recommendations from the existing system (Castells et al., 2021). However, metrics over top-$k$ items are difficult to optimize using supervised learning since the ranking operation is non-differentiable.

To address the mismatch between training objective and task goal, we propose a reinforcement learning (RL) framework that can optimize arbitrary task goals. Rather than training from scratch, we assume access to a supervised model that has been trained on user feedback data. Using the supervised model as the initial policy, we apply a policy gradient algorithm (Sutton et al., 1999) to finetune a model towards a task-specific goal like recall or precision at rank $k$ . Such an approach is made possible by recent advances in large language models (LLMs) that can provide accurate relevance judgments (He et al., 2023). As a result, LLMs can be used as a reward model to evaluate relevance of any top-k list generated during policy training. Compared to past work on using RL for information retrieval that trained policies from scratch (Zhu et al., 2022; Pan et al., 2019; Wei et al.,

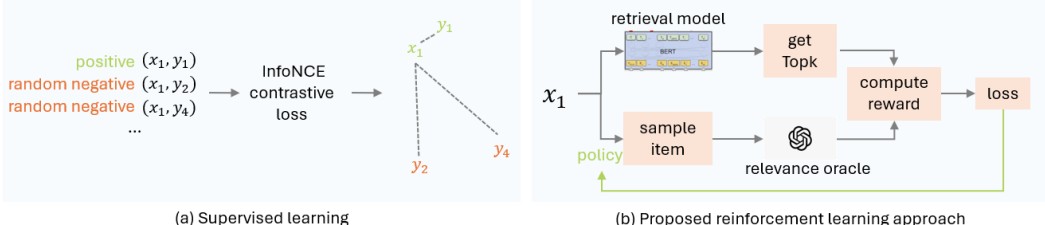

Figure 1: Supervised learning vs. the proposed reinforcement learning approach for retrieval model training. **Left:** InfoNCE contrastive loss with random negatives that does not consider the top-$k$ retrieved items or their optimality. **Right:** the proposed RL approach with explicit optimization for the top-$k$ retrieved items that can work for arbitrary task-specific losses.

2017), our insight is to use RL to finetune a pre-trained model and find solutions in its neighborhood of the initial policy (Pinto et al., 2023; Ouyang et al., 2022). We argue this is a feasible goal given the complexity of the retrieval task that often involves millions of states (input queries) and actions (items), and utilizes the complementary strengths of supervised learning and reinforcement learning.

Even when finetuning an existing policy, the large action space in retrieval tasks—equal to the number of available items that can be predicted for an input query—presents a challenge. While policy gradient methods do not explicitly parameterize the action space and are better suited for large action spaces, recent work shows that they are unable to work well beyond tens of items (Lu et al., 2022). These results are also supported by finite sample convergence results of policy gradient algorithms (Mei et al., 2020), which we extend to show a direct dependence of policy error on the square of the action space size.

To address the issue of optimizing over large action spaces, we propose a *simple* transformation of the problem: rather than considering a query as the state, we consider the (query, item) pair as a state. As a result, the action space is converted to a binary space: whether the item is preferred by the task-specific goal (e.g., relevance) or not. Theoretically, our formulation reduces the dependence of convergence error from quadratic to linear in the action space, which is a substantial reduction whenever the action space is large. Crucially, even though the action space is binary given a query-item pair, policy optimization can use rewards based on the top-$k$ retrieved list from the current policy, which was not possible in supervised learning. An added benefit is that since both query and item are a part of the state, we can add exploration to the policy gradient algorithm by sampling different items for each query, e.g., by using a separate, high-capacity retrieval model.

We evaluate the proposed algorithm, PG-Ret, on two datasets for the task of query-ad recommendation. The first dataset is a public query-ad dataset witth 4K actions while the second dataset is derived from a commercial search engine's logs and has 1.9M actions. On both datasets, we find that PG-Ret can improve the recall of top-k retrieved items compared to the supervised model. On the second dataset, we also consider a different task-specific goal: improving the novelty of the top-k items compared to an existing set of items. Here too PG-Ret improves upon the supervised model (obtaining a 45% increase in novelty) while incurring a marginal decrease in precision (relevance quality) of the top-$k$ list. Our technique, based on the policy gradient algorithms, is simple to implement (see Algorithm 1) and shows the benefit of finetuning models with RL for the desired metric after training with supervised learning.

To summarize, our contributions include,

- A simple, model-agnostic technique to finetune retrieval models that works for large action spaces (millions of actions) and arbitrary task-specific goals.

- Theoretical and empirical justification of the method across retrieval tasks, including an application to increase novelty of ad recommendation results on a commercial search engine.

## 2 RELATED WORK

Retrieval of top-k items can be considered as a one-step RL problem with large action space. We review the relevant literature in information retrieval, recent work in RL with large action spaces, and applications of LLMs to retrieval.

**RL and information retrieval.** Given a query, information retrieval can be divided into two phases; 1) retrieval of relevant items from a large candidate pool of items; 2) ranking the retrieved items to select a smaller set of items that are shown to the user. RL algorithms find many applications in the ranking phase, including contextual bandits (Li et al., 2010), markov decision processes (Wei et al., 2017), policy gradient algorithms (Pan et al., 2019; Xu et al., 2020) and off-policy methods (Chen et al., 2019). Many of these algorithms introduce a task-specific loss to the ranking problem. However, applying these techniques to the retrieval phase is challenging because of the large number of candidate items (actions). Recently, retrieval models are also used for selecting the in-context examples for including in an LLM's prompt and RL algorithms such as Q-learning (Zhang et al., 2022) and policy gradient (Lu et al., 2022) have been applied to the problem. However, policy gradient-based policy optimization does not work satisfactorily beyond a small number of actions ($\sim$20), as reported by (Lu et al., 2022). We aim to resolve this through our reparameterization of the policy gradient algorithm that reduces the large action space to binary actions.

**RL with large action spaces.** Large action spaces is a general challenge for RL beyond retrieval models (Dulac-Arnold et al., 2015), even when using policy gradient. For one-step RL problems, recent theoretical work in contextual bandits (Zhu et al., 2022; Lopez et al., 2021) tries to address the large actions problem. However, the focus of these approaches is on obtaining an optimal solution from scratch, which may be difficult in practice and misses the complementary benefits of supervised learning over user feedback. Instead, we focus on finetuning an existing supervised model, as proposed by Pinto et al. (2023) for computer vision models.

**LLMs for information retrieval.** Recently, LLMs like GPT-3.5 have been applied to retrieval tasks in a zero-shot setting with encouraging results (Dai et al., 2022; Hou et al., 2023). Instead of using compute-intensive LLMs directly for the retrieval task, here we aim to use LLMs as reward models to train an efficient, small retrieval model.

## 3 MOTIVATION AND PROBLEM STATEMENT

Commonly used retrieval models use a bi-encoder architecture (Gao et al., 2021), where the same neural network model embeds a query and item into a common representation space. The top-k items are selected based on the nearest neighbors to a query, as measured by a suitable distance function over the embeddings (e.g., cosine similarity). The encoder $\phi$ is typically optimized using a variant of contrastive learning, encouraging that positive <query,item> pairs in the training set should be closer in embedding space than non-positive pairs. Non-positive pairs may be random pairs or negative pairs labelled by user or mined from train data. Thus, given a train dataset $D \sim \mathcal{D}$ with positive query-item pairs, set of queries $X$ and items $Z$, and a similarity function $\mathrm{sim}$,

$$\hat{\phi} = \arg\min_{\phi} -log \sum_{(x,z)\in\mathcal{D}} \frac{\exp(sim(\phi(x),\phi(z)))}{\sum_{z'\in neg(x)}\exp(sim(\phi(x),\phi(z')))} \tag{1}$$

$$\mathbf{y}(x) = \mathrm{Topk}_{z\in Z}\, sim(\hat{\phi}(x),\hat{\phi}(z))$$

where $\mathbf{y} = [y_1, y_2, ..y_k]$ refers to the sequence of top-k items returned by a trained encoder $\hat{\phi}$ and $\mathrm{topk}$ is a non-differentiable ranking operation. Here we presented the InfoNCE contrastive loss (Oord et al., 2018) for training. At inference time, given a query, the trained encoder $\hat{\phi}$ is used to compute the similarity with each item and the top-k closest items are returned as the prediction. These top-k items are then evaluated on a task-specific metric. For example, for search and recommendation tasks, a common metric is recall@k or precision@k, corresponding to the quality of the top-k returned items. An additional metric of interest is novelty wrt. existing algorithms, especially in real-world systems with multiple deployed algorithms. Note that these task-specific metrics would not be optimized using supervised learning since these metrics are not differentiable.

In this paper, we develop a technique to finetune the supervised model for the task-specific metrics. Formally, the problem can be formulated as a one-step RL problem. In the standard formulation, the query is considered as the *state* and the set of top-k items as the *action*. A policy $\pi : X \to \{Z_k : Z_k \subseteq Z\}$ is a function that outputs a set of top-k items given a query. For each action selected by the policy, the environment provides reward feedback on the $<$ state,action $>$ pair. For instance, in the web search task, user's input query is the state, the top-k webpages are the action, the policy outputs top-k results given the query, and the environment provides reward based on the query and returned webpages. Given a reward function $\mathcal{R}$, the task to learn a policy $\pi_\theta$ (parameterized by $\theta$), that optimizes,

$$\max_\theta \mathbf{E}_{x \sim \mathcal{D}} \mathbf{E}_{\mathbf{y} \sim \pi_\theta(x)} \mathcal{R}(x, \mathbf{y}) \tag{2}$$

The ideal reward function for $\mathbf{y}$ is human feedback, but it is infeasible to obtain at scale. Therefore, for offline finetuning of the retrieval policy, the choice of a reward model is critical. Since LLMs have been shown to be competent at labelling retrieval outputs He et al. (2023), we utilize large language models (LLMs) to simulate human feedback.

## 4 LARGE ACTION POLICY GRADIENT

Since retrieval problems often involve a large number of states and actions, we use policy gradient algorithms to optimize the reward. As in past work Pinto et al. (2023); Lu et al. (2022), to avoid the complexity of predicting a *sequence* of actions, the policy is parameterized as a discrete action policy that outputs the probability of selecting each action (item) at a time, $\pi : X \to \{1, 2, 3...|Z|\}$. The top-k items, $\mathbf{y}$ are independently sampled from the discrete action probability distribution.

$$\pi_\theta : X \to Z; \ y_j \sim \pi_\theta(x) \forall j \in \{1, 2, 3..k\} \tag{3}$$

For example, if we use an encoder model, $f : X \cup Z \to \mathbf{R}^d$, then $\pi_\theta(z|x) = \mathrm{softmax}_Z \, sim(f(x), f(z))$. Then, given a reward function for top-k items, we approximate the expectation in Eqn. 2 using Monte Carlo sampling from the train set of queries.

$$\mathbf{E}_{x \sim \mathcal{D}} \mathbf{E}_{\mathbf{y} \sim \pi_\theta(x)} \mathcal{R}(x, \mathbf{y}) \approx \frac{1}{B} \sum_{i=1}^{B} \mathcal{R}(x^{(i)}, \mathbf{y}^{(i)}) \ \text{where} \ y_j^{(i)} \sim \pi_\theta(x^{(i)}) \ \forall j \in \{1, 2, 3..k\} \tag{4}$$

where B is the batch size. For simplicity, we use the REINFORCE algorithm (Williams, 1992) but we can use any other policy gradient algorithm here. The loss gradient is given by,

$$\nabla \mathbf{E}_{x \sim \mathcal{D}} \mathbf{E}_{\mathbf{y} \sim \pi_\theta(x)} \mathcal{R}(x, \mathbf{y}) = \mathbf{E}_{x \sim \mathcal{D}} \mathbf{E}_{\mathbf{y} \sim \pi_\theta(x)} \mathcal{R}(x, \mathbf{y}) \nabla \log \pi_\theta(\mathbf{y}|x)$$

$$\approx \frac{1}{B} \sum_{i=1}^{B} \mathcal{R}(x^{(i)}, \mathbf{y}^{(i)}) \nabla \log \pi_\theta(\mathbf{y^{(i)}}|x^{(i)}) \ \text{where} \ y_j^{(i)} \sim \pi_\theta(x^{(i)}) \ \forall j \in \{1, 2, 3..k\}$$

$$\tag{5}$$

Since the reward is one-step, the above optimization has a simple goal: increase the probability of items that occur in a k-sequence with high reward, and decrease the probability of items that occur in a k-sequence that obtains low reward. However, there are two challenges: **1)** the number of actions can large such that probability of sampling any single action may be close to zero in finite samples; **2)** the feedback is over the full k-sequence and may not map to the output semantics of $\pi_\theta$. For example, a k-sequence may obtain high reward due to some relevant items, but gradient would result in increasing the action probability of the irrelevant items in the k-sequence too.

### 4.1 THE CHALLENGE WITH LARGE ACTION SPACES

To address the above challenges due to the large action space, we turn to finite sample convergence bounds for the policy gradient algorithm. Extending the convergence results in Mei et al. (2020) for applying policy gradient to our setup, we show that error of a policy wrt. the optimal policy increases proportional to the square of the number of actions. Let $\pi^*$ denote the optimal policy that maximizes Eqn 2 and $t$ refer to the steps of the optimization.

**Assumption 1** (from Mei et al. (2020)). *(Sufficient exploration). The initial state $\mu$ distribution satisfies $\min_s \mu(s) > 0$.*

**Assumption 2.** *For each state, the initial policy's probability for selecting the optimal action is better than random (uniform probability) within some multiplicative constant $\rho \geq 1$: $\pi_{\theta_0}(\mathbf{y}^*(x)|x) > \frac{1}{\rho|Z|} \ \forall x \in D$ where $\mathbf{y}^*(x) := \arg\max_{\mathbf{y}} \pi^*(\mathbf{y}|x)$.*

**Proposition 1.** *Let Assumptions 1 and 2 hold and let $\{\theta_t\}_{t \geq 1}$ be generated using the standard policy gradient update: $\theta_{t+1} \leftarrow \theta_t + \eta \frac{\partial V^{\pi_{\theta_t}}(\mu)}{\partial \theta_t}$ where $V^{\pi_{\theta_t}}(\mu) = \mathbf{E}_{x \sim \mathcal{D}} \mathbf{E}_{\mathbf{y} \sim \pi_{\theta_t}(.|x)} \mathcal{R}(x, \mathbf{y})$. Then, for all $t \geq 1$, with $\eta = 1/8$,*

$$\mathbf{E}_{x \sim \mathcal{D}}[(\pi^*(x) - \pi_{\theta_t}(x))^T \mathbf{r}(x)] \leq \frac{16SA^2\rho^2}{t} \left\| \frac{1}{\mu} \right\|_\infty \tag{6}$$

*where $S = |X|$ is the number of states and $A = |Z|$ is the number of actions.*

Proof is in Appendix. The proof uses Thm. 4 from Mei et al. (2020) for markov decision processes and adapts it to the single-step problem and additionally uses Assumption 2 to express the bound in terms of $A = |Z|$. Note that the error in expected reward of a policy wrt. the optimal policy is proportional to the square of the number of actions.

### 4.2 REDUCTION TO BINARY ACTION SPACE

To improve the convergence rate, a naive solution may be filter the number of actions apriori (Lopez et al., 2021; Lu et al., 2022) using some criteria, but in practice it is difficult to obtain a principled criteria to filter the set of actions. Instead, we consider a different formulation where the state is $\langle query, item \rangle$ pair and the policy outputs the probability of selecting the item as a *relevant* item for the query (where "relevance" is defined according to the task-specific reward function). With this formulation, the number of states increases to $SA$ but the number of actions reduces to 2. As we show below, the convergence rate is significantly faster since the error grows linearly with A rather than quadratic.

**Proposition 2.** *With the new formulation, under the assumptions of Proposition 1, for all $t \geq 1$,*

$$\mathbf{E}_{x \sim \mathcal{D}}(\pi^*(x) - \pi_{\theta_t}(x))^T \mathbf{r}(x) \leq \frac{64SA\rho^2}{t} \left\| \frac{1}{\mu} \right\|_\infty \tag{7}$$

Note that in practice, $\rho$ may be higher for the binary action formulation since there are only 2 actions. Assuming a "good enough" supervised policy, conservative value for $\rho$ may be $\sim 50$, implying that probability of the optimal action under supervised policy is always $\geq 1/(2 \times 50) = 0.01$, whereas $\rho$ for Proposition 1 may be 1. Even then, as long as the number of actions is of the order of millions, $\rho^2 << A$ and hence the convergence rate in Proposition 2 would be significantly faster. In other words, as long as $\rho^2 << A$, the binary-action policy gradient algorithm will converge at a faster rate.

A key benefit of our formulation is that the reward can still be a function of the top-k retrieved items, even though the action is binary and conditioned on a query and item. This is because the environment can decide the reward based on whether the item is a part of the top-k items for the query retrieved by the current $\pi_{\theta_t}$. Specifically, given a state $\langle query, item \rangle$ and an action $a \in \{0, 1\}$, the reward is dependent on whether the *item* is part of the top-k items returned by $\pi_{\theta_t}$. For example, here is an example of a reward function for optimizing recall, given any query $x$ and item $z$, and a relevance oracle, Rel (e.g, an LLM).

$$\mathcal{R}_b(x, z) = \begin{cases} 1 & \text{if } \mathrm{Rel}(x, z) = 1 \text{ and } z \notin topk \\ -1 & \text{if } \mathrm{Rel}(x, z) = 0 \text{ and } z \in topk \\ 0 & \text{if } \mathrm{Rel}(x, z) = 1 \text{ and } z \in topk \\ 0 & \text{if } \mathrm{Rel}(x, z) = 0 \text{ and } z \notin topk \end{cases} \tag{8}$$

The full reward function is given by, $\mathcal{R}((x, z), a) = a\mathcal{R}_b(x, z) + (1 - a)(-\mathcal{R}_b(x, z))$. Broadly speaking, if a relevant item is not among the top-k items, then its action probability should be increased. And if an irrelevant item is among the top-k items, then its action probability should be decreased. The corresponding gradient is,

$$\nabla \mathbf{E}_{x,z \sim \mathcal{D}} \mathbf{E}_{\mathbf{a} \sim \pi'_\theta(x,z)} \mathcal{R}((x, z), a)$$

$$\approx \sum_{i=1}^{B} [a^{(i)} \mathcal{R}_b(x^{(i)}, z^{(i)}) - (1 - a^{(i)}) \mathcal{R}_b(x^{(i)}, z^{(i)})] \nabla \log \pi'_\theta(a^{(i)} | x^{(i)}, z^{(i)}) \tag{9}$$

where $\pi'(a = 1|x, z) = sim(f(x), f(z))$ and $\pi'(a = 0|x, z) = 1 - sim(f(x), f(z))$. We use a similarity function that is within $[0, 1]$. Coming back to challenge **2)** in the original formulation, here the loss gradient would not increase the $\langle query, item \rangle$ score ($\pi'(a = 1|x, z)$) if the item is non-relevant but is somehow included in top-k results. The score $\pi'(a = 1|x, z)$ is increased only when $z$ is relevant for $x$, as measured by a relevance oracle. This addresses challenge 2) in the original formulation.

## 4.3 PROPOSED ALGORITHM: PG-RET

The above discussion indicates the following conditions for a fast convergence rate with policy gradient, **1)** the action space should be small; and **2)** the initial policy should be as close to the optimal as possible. We addressed the first condition through a binary-action formulation of the problem. For the second condition, we propose initializing the policy optimization with a policy trained using supervised learning (e.g., using Eq. 1).

Since items are also a part of the state in our proposed formulation, a key outstanding question is how to sample the states $\langle x, z \rangle$. To map to the original policy gradient formulation, we can sample queries $x$ randomly from the train dataset. For each query, we compute the similarity scores with all items and then sample $k$ items proportional to their score. Note that we are not restricted to only sampling items proportional to their similarity score (since items are not actions now). Therefore, we also add exploration by sampling top-k $(x, z')$ items from another retrieval model (trained independently on separate training data). For example, for query-item text relevance, we may use any of the pre-trained bi-encoders such as BERT or gtr-t5 models. Finally, there is a risk that the policy overfits to the reward from the relevance oracle and "forgets" the true user feedback data on which the supervised policy was trained. Thus, we also add a random sample of the user feedback data $(x, z_{user})$. To combine the three sources, we sample x randomly from the dataset and then sample $z \sim \pi_{\theta_t}(x)$ with probability $\alpha$; $z \sim \pi_{\theta explore_t}(x)$ with probability $\beta$; or $z \sim P_{\mathcal{D}}(x)$ with probability $1 - \alpha - \beta$, with a budget of $k$ item samples.

Note that to compute top-k items for a sampled query, the entire set of actions have to be encoded by $f$. For computationally efficiency, we fix the item encoder to be the initial encoder after supervised training. This avoids having to recompute the item embeddings for each query in the batch. That is, only the query encoder is updated during policy optimization. The full algorithm is shown in Algorithm 1.

---

**Algorithm 1** Large action PG

1: **Input:** Initial policy $\pi_\theta$, Training dataset $D$, Reward function $\mathcal{R}$, Number of epochs $N$, Batch size $B$, Learning rate $\eta$, $\alpha, \beta \in [0, 1]$
2: **Output:** Trained policy $\hat{\pi}_\theta$.
3: **for** epoch=1,2,3..N **do**
4:     **for** $D_{batch} \sim D$ **do**
        $L = 0$
        $i = 0$
        $(x, z) \sim D$
        **while** $i < B$ **do**
        $a \sim \pi'_\theta(x, z)$ // Sample Action
        $A_k = \text{topk}_Z(x; \pi_\theta)$ // Top-k items
        $r_b = \mathcal{R}_b(x, z, A_k, \text{rel}(x, z))$ // Reward depends on top-k items and relevance oracle
        $r = [r_b a + (-r_b)(1 - a)] \log \pi'_\theta(a|x, z)$
        $L = L - r$
        $i = i + 1$
        $x \sim D$
        $z \sim \pi_\theta(x)(\text{wProb } \alpha), \sim \pi_{\theta explore}(x)(\text{wProb } \beta), \text{ or } \sim D(\text{wProb } 1 - \alpha - \beta)$
6:     **end while**
        $\theta = \theta - \eta \nabla L$ // Can use any gradient optimizer
7:     **end for**
8: **end for**

---

| Dataset | Train inputs | Test inputs | Number of Actions |
|---|---|---|---|
| QADSM | 100K | 10K | 4K |
| Query-Keyword | 110K | 4.2K | 1.95M |

Table 1: Dataset statistics for QADSM (public dataset) and Query-Keyword dataset from a commercial search engine. The Query-Keyword dataset has over 1.9M actions.

## 5 EVALUATION

We evaluate PG-Ret on top-k retrieval for ad recommendation with two task-specific goals: recall of the top-k retrieved items and novelty of the retrieved items wrt. existing algorithms' output.

**Datasets.** We use a public dataset and dataset from a commercial search engine (see Table 1).

- **QADSM.** This dataset is part of the X-GLUE benchmark Liang et al. (2020) and contains query-ad pairs from three different languages. For simplicity, we consider the query-ad pairs in English. The training set consists of 100K query-ad pairs and test set consists of 10K query-ad pairs. We consider Recall@K as the task-specific metric. We use gtr-t5-base as the reward model for this task since it obtains reasonably high accuracy on this dataset. For recall evaluation, we only consider the positive query-ad pairs from the test set.

- **Query-keyword recommendation**. This dataset contains queries and ad keywords from a commercial search engine's logs. The goal is to produce novel ad keywords that are not produced by the production algorithm trained on click data. To do so, we collect new relevance data using GPT-4. A supervised model is finetuned using this data. After that, we apply PG-Ret to increase novelty while retaining relevance. In addition, we also consider the standard goal of increasing Recall for the retriever. We use GPT-3.5 as the reward model with a prompt designed to capture whether the query and ad keyword have the same intent.

**Baselines.** For each dataset, we compare PG-Ret to the base model and the model trained using supervised learning on the training data. We also compare to the reward model for QADSM since gtr-t5-base can also be used as an (inefficient) bi-encoder retriever. All models were trained using Adam optimizer with a learning rate of $10^{-4}$ and batch size of 128.

### 5.1 OPTIMIZING RECALL ON QADSM DATASET

We use a base model pretrained on a semantic similarity task, *stsb-xlm-r-multilingual*. We train a supervised model using InfoNCE contrastive loss and random negatives. Using the supervised model as initial policy, PG-Ret is trained to optimize recall directly using the reward function from Eqn 8. Table 2 shows the recall metric at two different values of k. As expected, supervised learning increases the accuracy of prediction. PG-Ret provides substantial gains on top of the supervised policy. For comparison, we also show the recall of "gtr-t5-base", the reward model in the experiment. Gtr-t5-base obtains the highest recall but at the cost of computational inefficiency. While PG-Ret does not reach the reward model's accuracy, it significantly improves upon the supervised learning model.

| Model | Recall@1 | Recall@10 |
|---|---|---|
| Base | 20.2 | 51.0 |
| Supervised | 22.1 | 52.3 |
| PG-Ret | 25.0 | 56.9 |
| Reward Model | 29.3 | 62.1 |

Table 2: Recall metric for different models on the QADSM dataset. PG-Ret improves upon the recall obtained by the supervised model.

| Model | SimCSE(R@1) | SimCSE(R@10) | SimCSE-Ads(R@1) | SimCSE-Ads(R@10) |
|---|---|---|---|---|
| Pretrained | 15.4 | 47.1 | 21.7 | 68.4 |
| Supervised | 18.3 | 57.7 | 22.4 | 70.1 |
| PG-Ret | **24.6** | **62.6** | **26.9** | **71.2** |

Table 3: Recall@1 and Recall@10 for different algorithms on the Query-Keyword retrieval task. The first two columns use the public SimCSE checkpoint whereas the last two use a checkpoint pretrained on search engine data. PG-Ret obtains substantially higher recall@1 for both base models.

| Model | Novelty@10 | Precision@1 | Precision@5 |
|---|---|---|---|
| Pretrained | - | 86.3 | **74.9** |
| Supervised | 2.9 | 86.5 | 74.5 |
| PG-Ret | **4.2** | **88.7** | 72.5 |

Table 4: Novelty@10 (wrt the pretrained model) and Precision@1 for different algorithms on the Query-Keyword retrieval task. Top-10 keyword recommendations from PG-Ret yield 45% higher diversity than the supervised model. In addition, the drop in precision due to higher diversity is minimal (4% drop in Precision@5).

## 5.2 Optimizing recall and novelty on Query-keyword recommendation task

For the query-keyword dataset, the goal is to increase the novelty of policy's recommendations. As a warmup, we first consider increasing Recall of the finetuned model. The reward function used is the same as Eq.8 with the relevance function being GPT-3.5. That is, for determining query-ad relevance, we consider GPT-3.5 as the oracle and design a suitable prompt asking whether the query and keyword share the same intent. We set k=2, since we are focused on the task of improving recall over top-ranked items.

For this task, we consider SimCSE Gao et al. (2021) as the base model. We consider two versions of pretrained SimCSE: the publicly available one and a custom SimCSE model trained on search engine data. For each pretrained version, we compare PG-Ret to a model learnt using supervised learning. As Table 3 shows, supervised models increase Recall@K compared to the pretrained base model. However the gains due to PG-Ret are substantially higher. In particular, recall@1 increases substantially compared to both supervised models.

We next optimize for novelty and add a new constraint to the reward model. In addition to relevance using GPT-3.5, we also consider whether the keyword is already in the top-k list of keywords returned by the production algorithm. If it is, it is considered "irrelevant" and its action probability should decrease, otherwise we follow GPT's relevance label.

Using this modified reward, we obtain a policy that has significantly more novel keywords compared to the supervised policy. For calculating diversity, we measure the number of novel keywords that are returned at rank k=10 compared to the base model. The results are shown in Table 4. To ensure that relevance of the new keywords is not reduced, we also report the precision of the top-5 keywords returned by PG-Ret. For evaluating precision, the relevance is calculated using GPT-4 (to avoid overfitting issues due to the relevance oracle being GPT-3.5). We observe that the drop in precision is marginal (4%) compared to the increase in novel keywords returned (45% increase compared to the supervised model).

Finally, we compare PG-Ret on real user traffic using an A/B experiment. Keywords from PG-Ret are selected for impressions only if they are novel compared to the existing algorithm. We observe a 0.2% statistically significant increase in click yield on a 14 day period.

## 6 Limitations and Conclusion

We presented a technique to optimize a non-differentiable, task-specific loss in information retrieval applications. We justified the binary-action formulation of the problem through theoretical and

empirical results. On two recommendation datasets, the proposed technique leads to substantial gains in the task-specific metric.

That said, our work has limitations. We consider a simple REINFORCE-based policy gradient algorithm. Future work can consider Actor-critic or PPO algorithms for optimizing task-specific rewards. Moreover, based on Proposition 2, our method is expected to work only when the initial supervised policy is close to the optimal policy. Building policy optimization algorithms with principled exploration in large action spaces is an open research question.

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

## A APPENDIX: PROOF OF PROPOSITIONS

### A.1 PROPOSITION 1

To prove Proposition 1, we restate a result from Mei et al. (2020) for a MDP. Here $\gamma$ is the discount factor, $t$ refers to steps of the optimization, and $d_\mu^\pi(\gamma) = \mathbf{E}_{s_0 \sim \mu}(1-\gamma)\sum_{t=0}^\infty \gamma^t P(s_t = s | s_0, \pi)$ is the discounted state distribution.

**Theorem 1** (Mei et al. (2020)). *Let Assumption 1 hold and let $\{\theta_t\}_{t \geq 1}$ be generated using $\theta_{t+1} \leftarrow \theta_t + \eta \frac{\partial V^{\pi_{\theta_t}}(\mu)}{\partial \theta_t}$ where $V^{\pi_{\theta_t}}(\mu) = \mathbf{E}_{s \sim \mu} V^{\pi_{\theta_t}}(s)$. Let $\eta = (1-\gamma)3/8$, $c$ be the positive constant $c := \inf_{s \in S, t \geq 1} \pi_{\theta_t}(a^*(s)|s) > 0$. Then, for all $t \geq 1$,*

$$\mathbf{E}_{s \sim \mu}[V^*(s) - V^{\pi_{\theta_t}}(s)] \leq \frac{16S}{c^2(1-\gamma)^6 t} \left\| \frac{d_\mu^{\pi^*}(\gamma)}{\mu} \right\|_\infty^2 \left\| \frac{1}{\mu} \right\|_\infty \tag{10}$$

**Proposition 1.** *Let Assumptions 1 and 2 hold and let $\{\theta_t\}_{t \geq 1}$ be generated using the standard policy gradient update: $\theta_{t+1} \leftarrow \theta_t + \eta \frac{\partial V^{\pi_{\theta_t}}(\mu)}{\partial \theta_t}$ where $V^{\pi_{\theta_t}}(\mu) = \mathbf{E}_{x \sim \mathcal{D}} \mathbf{E}_{\mathbf{y} \sim \pi_{\theta_t}(.|x)} \mathcal{R}(x, \mathbf{y})$. Then, for all $t \geq 1$, with $\eta = 1/8$,*

$$\mathbf{E}_{x \sim \mathcal{D}}[(\pi^*(x) - \pi_{\theta_t}(x))^T \mathbf{r}(x)] \leq \frac{16SA^2\rho^2}{t} \left\| \frac{1}{\mu} \right\|_\infty \tag{6}$$

*where $S = |X|$ is the number of states and $A = |Z|$ is the number of actions.*

*Proof.* Note that our RL setup is one-step, hence we can assume $\gamma = 0$. Then $V^\pi(s)$ from Theorem 1 simplifies to,

$$V^\pi(s) = \mathbf{E}_{a_t \sim \pi(.|s_t)} \sum_{t=0}^\infty \gamma^t r(s_t, a_t) = \mathbf{E}_{a_t \sim \pi(.|s_t)} r(s_0, a_0) = \pi(s)^T \mathbf{r}(s)$$

where $\pi$ and $r$ in the last equation refer to the vectors of action probabilities and their rewards given a state. Further, since $\gamma = 0$, $d_\mu^{\pi^*}(\gamma) = \mu$. Hence, we can write Theorem 1 as,

$$\mathbf{E}_{s \sim \mu}[V^*(s) - V^{\pi_{\theta_t}}(s)] = \mathbf{E}_{s \sim \mu}[\pi^*(s)^T \mathbf{r}(s) - \pi_{\theta_t}(s)^T \mathbf{r}(s)] \tag{11}$$

$$= \mathbf{E}_{s \sim \mu}[(\pi^*(s) - \pi_{\theta_t}(s))^T \mathbf{r}(s)] \tag{12}$$

$$= \mathbf{E}_{x \sim \mathcal{D}}[(\pi^*(x) - \pi_{\theta_t}(x))^T \mathbf{r}(x)] \tag{13}$$

$$\leq \frac{16S}{c^2(1-\gamma)^6 t} \left\| \frac{d_\mu^{\pi^*}(\gamma)}{\mu} \right\|_\infty^2 \left\| \frac{1}{\mu} \right\|_\infty \tag{14}$$

$$\leq \frac{16S}{c^2 t} \left\| \frac{1}{\mu} \right\|_\infty \tag{15}$$

where the third inequality is because the initial state distribution is the distribution of queries in the training data.

Now, using Assumption 2, the minimum initial probability for the optimal action $a^*$ is $\frac{1}{\rho A}$ for all states. Assuming that the gradient updates do not decrease the probability of the optimal action, $c = \inf_{s \in S, t \geq 1} \pi_{\theta_t}(a^*(s)|s) = \frac{1}{\rho A}$. Substituting c in the above equation, we obtain the result.

$$\mathbf{E}_{x \sim \mathcal{D}}[(\pi^*(x) - \pi_{\theta_t}(x))^T \mathbf{r}(x)] \leq \frac{16SA^2\rho^2}{t} \left\| \frac{1}{\mu} \right\|_\infty$$

$\square$

### A.2 PROPOSITION 2

**Proposition 2.** *With the new formulation, under the assumptions of Proposition 1, for all $t \geq 1$,*

$$\mathbf{E}_{x \sim \mathcal{D}}(\pi^*(x) - \pi_{\theta_t}(x))^T \mathbf{r}(x) \leq \frac{64SA\rho^2}{t} \left\| \frac{1}{\mu} \right\|_\infty \tag{7}$$

*Proof.* Using the equation from Proposition 1 and substituting $S = SA$ and $A = 2$ leads us to the result. $\qquad\square$

