# OpenReview forum: "Adapting Retrieval Models to Task-Specific Goals using Reinforcement Learning"
_ICLR.cc/2024/Conference — Submitted to ICLR 2024_

### Official Review · Reviewer_MtVy · 2023-10-28

**Soundness:** 2 fair
**Presentation:** 3 good
**Contribution:** 2 fair
**Rating:** 3
**Confidence:** 3

**Summary:**

The submission studies the problem to optimize task-specific metrics in retrieval systems via reinforcement learning. The topic itself is standard and has a lot of literature. The main contribution of the paper is the proposal to treat query-item as the state which can address some disadvantages of large action space. Some theoretical analysis is provided. The paper also use LLMs as the reward model. For experiments, one public dataset and one internal dataset is used. Basic baselines are compared against and the proposed method shows some performance benefits.

**Strengths:**

S1: Though RL for information retrieval has a rich literature, the formulation to model query-item as the state looks interesting to the reviewer, though the reviewer does not have the expertise to comment on the theory part.

S2: It is interesting to see two different objectives, including recall and a novelty metric.

**Weaknesses:**

W1: The experiments are quite weak and non-standard. This weakness itself may warrant rejection in a top venue. There are numerous retrieval datasets and strong baselines and it is not clear why the authors selected the datasets (1 public dataset that is not commonly used) and an internal dataset, and pretty much without any sensible baselines. Especially for the recall metric, it is standard so it’s not clear why no standard baselines or datasets are used. The internal dataset does not add much value to the paper, as the details are unclear, will not help reproducibility, and the real-world impact is not clear given no online experiments. There are many ways to compose a task that do not optimize recall metric so it is not clear why such task is chosen. The authors implemented the basic base models themselves. The choices also look arbitrary, such as the choice of certain model architectures, the LLMs used (while assuming they are reliable which is not the case - despite recent papers on the popular topic, they hardly beat previous tuning methods). All these will make the impact of the work hard to be measured and the reproducibility of the proposed method extremely difficult.

W2: RL in information retrieval has a very rich literature, and optimizing arbitrary metric is inherited from RL, so the novelty/story from this perspective is limited. The major contribution is really the algorithm mentioned in S1, but the significance of the proposed method is unclear.

**Questions:**

See weaknesses.

How valid is the assumptions made on LLMs? Using LLMs as rater is a promising area, but not solved problem.

---

> ### Author Response · Authors · 2023-11-18
>
> Thank you for your comments. Answering specific questions below.
>
> 1. Our goal was to choose a dataset where LLM can be reliably used as a reward model. That is why we chose symmetric short-text retrieval tasks where the relevance function can be described easily using LLM.
> 2. The novelty is the application to large action spaces. Most RL work in information retrieval focuses on tens of actions, whereas our method can scale to millions of actions.
> 3. Using LLM as a rater requires that the prompted LLM align with the task’s objectives. That is why we choose symmetric, short-text retrieval tasks where the LLM can be reliably used.

---

### Official Review · Reviewer_ACSz · 2023-10-31

**Soundness:** 2 fair
**Presentation:** 2 fair
**Contribution:** 2 fair
**Rating:** 3
**Confidence:** 2

**Summary:**

The paper tackles the problem of using an oracle (in this case an LLM) to augment an existing dataset for the purpose of learning to rank items, particularly for tasks where the optimal ranking for a given query is not as obvious as the top K items according to their relevance (for instance, when novelty is a criteria). The paper proposes using an LLM as a reward oracle for an additional reinforcement learning stage that would fine tune a model trained on labelled data, and further aligns the ranking model with the task at hand, more so than it would be able to learn from just the offline training data.

The performance of the new algorithm is highlighted on two datasets,  one publicly available, the other proprietary. The experiments show the fine-tuned algorithm exceeding the performance of the supervised model, but falling short of reaching the performance of the model used as an oracle (in this case the LLM)

**Strengths:**

The paper is fairly well written and tackles a relevant practical and widespread problem: misalignment between the learning to rank objective and the objectives supervised learning algorithms can actually use for training (which need to be differentiable).

The paper provides experiments on real world data and on open datasets (not just proprietary datasets).

**Weaknesses:**

I do not believe the significance of the approach presented here is substantial enough to warrant acceptance into the venue. Fine-tuning supervised models with a reinforcement learning phase is a well-known approach. Once it has been established that LLMs are suitable for labelling data for this sort of problems relatively reliably, using RL for fine-tuning does not feel like a novel contribution. It is also unclear how good the LLMs are at aligning the fine-tuning objective to the objective we are actually aiming to solve. For instance, if the criteria is diversity etc.

The reframing of the problem as having two action spaces is a bit unclear to me as to how it alleviates the complexity of the setting. I also fail to see how the problem is still cast as an MDP for the theoretical results to hold. This should be clearly articulated in the main body of the paper.

**Questions:**

Once we have an oracle, potentially other avenues of improving the performance appear. For instance, framing the problem as a bandit problem (for example contextual linear cascading bandit [1,2] where item vectors can be generated with the LLMs), considering the problem an example of Positive-Unlabelled [3] or Active learning, where we can augment the initial dataset based on weaknesses the model uncovers in its own predictions. In light of all these possible alternatives, what are the reason to believe using the REINFORCE algorithm for the fine-tuning step is an impactful approach and not just another approach?

Can you better describe the intuition why the convergence speed of the algorithm is substantially increased from framing the problem as having the states being pairs <state, item> and having binary actions?

In addition to the above explanation, I would also like to see a detailed formal description of the resulting MDP and the application of the Theorem in Mei et al. (2020) to the resulting setting. It is unclear to me what this MDP would look like and how the theorem applies.

Can you provide an interpretation of how impactful the $0.2\\%$ increase in the click yield is?

[1] - https://arxiv.org/abs/1502.02763
[2] - https://proceedings.mlr.press/v115/hiranandani20a/hiranandani20a.pdf
[3] - https://link.springer.com/article/10.1007/s10994-020-05877-5

---

> ### Author Response · Authors · 2023-11-18
>
> The main contribution of the work is to apply RL finetuning to large action spaces, which is a specific, hard application of RL.
>
> 1. Compared to positive-unlabelled or active learning setting, the benefit of RL is that we can optimize any non-differentiable loss. Contextual Bandits do not scale well to millions of actions. Most such approaches optimize over tens of items whereas our problem considers optimization over millions of items.
>
> 2. The intuition is that the convergence depends quadratically on number of actions and linearly in the number of states. So, our setup reduces the actions while increasing the number of states; the net result is a substantial increase in convergence speed.
>
> 3. It is a one-step MDP. In each round, a state (<query, item>) is sampled, an action is taken (0/1), and the environment provides the reward. It can be understood as a contextual bandit setup.
>
> 4. The click yield is an incremental metric: it is measured only for the novel keywords that are introduced by our algorithm but not produced by the production algorithm. The click yield is lower than our expectation but still significant.

---

> > ### Comment · Reviewer_ACSz · 2023-11-20
> > **Thank you for the clarifications**
> >
> > I would like a bit further clarification regarding point 2: In your setup you are gauging the relevance of individual items to the query as a proxy for its attractiveness in the presented ranking but this would not solve the issues of contrasting items in the list (like when diversity is a requirement of the resulting list). So when the reward of the ranking relies on dependencies between the items in the ranking, this setup would not solve the original alignment issue, correct?

---

> ### Author Response · Authors · 2023-11-21
>
> Yes, that is correct. Our formulation works whenever the reward for a list of items can be decomposed into a sum/aggregation of individual rewards for each item in the list (e.g., for the novelty reward w.r.t. a pre-existing top-k list). It will not work for diversity reward where two items in the predicted list need to be compared.

---

### Official Review · Reviewer_2vic · 2023-10-31

**Soundness:** 3 good
**Presentation:** 2 fair
**Contribution:** 3 good
**Rating:** 5
**Confidence:** 4

**Summary:**

* This paper studies the problem of optimizing retrieval models such that it optimizes for a more direct goal rather than typical self-supervised genetic objectives.
* Since the downstream performance is not differentiable and annotation ground-truth data is limited, the approach is based on using LLM reward estimators or evaluators to generate supervision signal.
* The idea is to build model to assess relevance of a certain item to context i.e. binary action space, rather than much larger space of ranking relevant items given a query; therefore it is more approachable from the RL learning and LLM reasoning perspective.

-- No change in evaluation after reading the authors responses

**Strengths:**

- The method is scientifically sound, intuitive, and useful for real-world applications

**Weaknesses:**

1- The discussion of large-action policy gradient in Section 4 (especially before 4.1) can be summarized or moved to appendix. It is a baseline but not the proposed method here. I think the saved space is better utilized if we could discuss the reward modeling method more clearly (e.g. how is it prompted, any key findings, etc).

2- Section 4.3 and Algorithm 1 is not clear. For example, it is not clear where the relevance oracle is coming from is it small set of annotations, or an LLM reward/relevance estimator?

3- For QADSM, the reward model seems rather small, especially compared to typical embedding models used for the retrieval. My understanding is that the reward model can/should actually be orders of magnitude larger and more capable to generate best supervision signal for the embedding model training. Any reason authors decided to use a variation of T5-base?

4- On the same topic, I see GPY-4/3.5 is used for the other dataset, isn’t it more intuitive to finetune an LLM for the specific relevance estimation task? any results/experiments to support using a strong but generic model?

**Questions:**

(see above for more questions)

About the usecase, I was wondering if authors could share any findings/experiments or thinking on how to apply such technique when new items are being added to the index. It is easy for an offline fixed dataset to build such relevance models or finetune embeddings but how would this work when new items are added? Do we need to retrain the models each time or we are claiming generalization in the learned embeddings

**Details Of Ethics Concerns:**

No particular concern

---

> ### Author Response · Authors · 2023-11-18
>
> Thanks for your valuable feedback. Answering specific questions below.
>
> 1. The relevance oracle is assumed to be a LLM reward estimator.
> 2. We use t5-base since its accuracy was already good on the dataset. It may be even more powerful to use a bigger model.
> 3. Agree. It will be ideal to finetune an LLM for each task. However, we found that GPT-3.5 provides good accuracy through a simple prompt. For QADSM, we could have used GPT-3.5 too but found that a t5-base model gave reasonable results too. So we choose t5-base to demonstrate that our method is not reliant on (costly) access to GPT-3.5.
> 4. The learned embedding model will have generalization. As long as the model to encode keywords is fixed, the same trained model for queries can be used.  In other words, our method, PGRetrieve, can be considered as an *offline RL* training method. Once trained, the model is used in an identical fashion to a model learnt using supervised learning.

---

> > ### Comment · Reviewer_2vic · 2023-11-21
> > **Re: Official Comment by Authors**
> >
> > Thank you for the response. While I appreciate the authors effort to answer my questions, my concerns are not addressed and data points and stronger rationale is expected. I would like to keep my evaluation as is.

---

### Official Review · Reviewer_mWCM · 2023-11-06

**Soundness:** 2 fair
**Presentation:** 3 good
**Contribution:** 3 good
**Rating:** 5
**Confidence:** 3

**Summary:**

In this paper, the authors propose a reinforcement learning method to fine tune an existing bi-encoder retrieval models. The proposed method, PG-Ret, considers query-document pair as state and binary [relevant/not-relevant] action space. The lower bound of convergence rate is given and the reduction of action space can have higher convergence rate. Empirical analysis on the QADSM public dataset and a keyword recommendation e-commerce private dataset show PG-Ret can improve recall and top-k diversity than the original bi-encoder model.

**Strengths:**

* The paper presents a novel method to conduct task-specific fine-tuning of embedding based retrieval models. The presentation of the paper is clear and easy to follow.
* The theoretical analysis seems correct and justifies the reduction of the action space.
* Empirical studies show the proposed method can improve the retrieval models without fine-tuning.

**Weaknesses:**

* The author claims the proposed method is applicable to general retrieval models yet only the InfoNCE with random negatives are used as baseline in the empirical study. It is know the negative sampling plays a very important role in the retrieval model training. Therefore the beselline supervised model could have a significant gain if the author train the supervised model with the three sources of positive/negative samples described in section 4.3 paragraph 2.
* The datasets seems to be toy-sized. More empirical results is needed to show the method actually works. Please consider add comparisons with SOTA retrieval method on more widely used retrieval benchmarks.
* The paper utilizes other pre-trained LLM models as relevance oracle which introduces additional supervision. I am not sure if the gain in recall comes from the reinforcement fine-tuning or simply from getting more supervision from a stronger model. Please consider add ablation study to validate the contribution from each part.
* From section 4.3, seems it is required to compute all query-document pairs in order to sample the states. This is computationally intensive and probably infeasible for most real applications. The space to store such dense query-document score matrix could be astronomical. That being said, I don't think this method can scale to real e-commerce applications as the author claims.

**Questions:**

* For the keyword recommendation experiment, why is the k set to be very small? That seems to be too small a match set to be judged for diversity.
* What's the prompt to get relevance judgement from GPT-3.5?
* How does the method improve upon SOTA retrieval models on other widely used retrieval benchmarks, such as MSMARCO/NQ?

---

> ### Author Response · Authors · 2023-11-18
> **Response to reviewer**
>
> Thanks for your feedback on the paper. Answering specific questions below.
>
> 1. We experimented with different values of k. Since the policy is an encoder model, we found that giving feedback on the topmost ranked items had a larger effect on the gradient than giving feedback on the other items. Therefore, to speed up training, we choose a small k. Note that changing the predictions at small k for a query also imply that predictions at larger k will change, since the encoder model provides a single representation of the query (and the representations for keywords are fixed).
>
> 2. Here’s the prompt used. 'Given the query “q”, is the following query expressing the exact same intent, "k"? Please answer in a single word: Yes/No.' Note that keyword is also presented as a query to the large language model.
> 3. Our focus is on symmetric, short-text retrieval tasks because our method depends on having a reward function that is aligned to the dataset’s distribution and task. MSMarco/NQ are more general passage-retrieval or question-answering tasks where novelty may have limited value.

---

### Meta-Review · Area_Chair_r7ty · 2023-12-10

**Metareview:**

I recommend to reject this paper.

   In this paper, the authors proposed to use a reinforcement learning approach (policy gradient) to adapt an existing bi-encoder retrieval model to optimize a more direct goal rather than typical self-supervised genetic objectives. Experimental results are conducted using one public dataset and one private dataset.

    I shared the similar comments from all reviewers that this paper does not meet the acceptance threshold for ICLR as the experiments are quite weak and non-standard. Thus I recommend to reject this paper.

**Justification For Why Not Higher Score:**

I agreed with all the reviewers that this paper is below the acceptance bar for ICLR due to the weakness of experiments. N.N/

**Justification For Why Not Lower Score:**

N/A

---

### Decision · Program_Chairs · 2024-01-16

Reject